# Novel Findings about Double-Loaded Curcumin-in-HPβcyclodextrin-in Liposomes: Effects on the Lipid Bilayer and Drug Release

**DOI:** 10.3390/pharmaceutics10040256

**Published:** 2018-12-03

**Authors:** Ana-María Fernández-Romero, Francesca Maestrelli, Paola Angela Mura, Antonio María Rabasco, María Luisa González-Rodríguez

**Affiliations:** 1Department of Pharmacy and Pharmaceutical Technology, Faculty of Pharmacy, Universidad de Sevilla, 41012 Seville, Spain; anaferrom2@alum.us.es (A.-M.F.-R.); amra@us.es (A.M.R.); 2Department of Chemistry, University of Florence, via Schiff 6, Sesto Fiorentino, 50019 Florence, Italy; francesca.maestrelli@unifi.it (F.M.); paola.mura@unifi.it (P.A.M.)

**Keywords:** liposome, curcumin, cyclodextrin, double-loading, deformability, antioxidant activity

## Abstract

In this study, the encapsulation of curcumin (Cur) in “drug-in-cyclodextrin-in-liposomes (DCL)” by following the double-loading technique (DL) was proposed, giving rise to DCL–DL. The aim was to analyze the effect of cyclodextrin (CD) on the physicochemical, stability, and drug-release properties of liposomes. After selecting didodecyldimethylammonium bromide (DDAB) as the cationic lipid, DCL–DL was formulated by adding 2-hydroxypropyl-α/β/γ-CD (HPβCD)–Cur complexes into the aqueous phase. A competitive effect of cholesterol (Cho) for the CD cavity was found, so cholesteryl hemisuccinate (Chems) was used. The optimal composition of the DCL–DL bilayer was obtained by applying Taguchi methodology and regression analysis. Vesicles showed a lower drug encapsulation efficiency compared to conventional liposomes (CL) and CL containing HPβCD in the aqueous phase. However, the presence of HPβCD significantly increased vesicle deformability and Cur antioxidant activity over time. In addition, drug release profiles showed a sustained release after an initial burst effect, fitting to the Korsmeyer-Peppas kinetic model. Moreover, a direct correlation between the area under the curve (AUC) of dissolution profiles and flexibility of liposomes was obtained. It can be concluded that these “drug-in-cyclodextrin-in-deformable” liposomes in the presence of HPβCD may be a promising carrier for increasing the entrapment efficiency and stability of Cur without compromising the integrity of the liposome bilayer.

## 1. Introduction

Curcumin (Cur) is the main active component in rhizomes of *Curcuma longa*, which is an ancient Chinese spice that is extremely appreciated in traditional medicine remedies [1]. It has been recently discovered that this compound has multiple applications as an anti-inflammatory, anti-cancer, antioxidant, anti-viral and cytoprotective agent [2]. As a polyphenol, Cur is capable of forming hydrogen bonds with other Cur molecules and not with water, which reduces its solubility in this medium [3]. Due to this problem, Cur exhibits a low water solubility, which implies low bioavailability [2,4,5] To overcome these drawbacks, numerous strategies have been proposed, such as cyclodextrin complexation [4,6], or entrapment in liposomes [7,8], micelles [9,10], dendrimers [11], etc.

Cyclodextrins (CD) are cyclic oligosaccharides containing a different number of units of (α-1,4)-linked-d-glucopyranoside. Modified CD, which is derived from native molecules (α-CD, β-CD, and γ-CD), was designed to increase its solubility. Some of these derivatives are 2-hydroxypropyl-α/β/γ-CD (HPβCD), methylated-β-CD (MβCD), sulfobutylether-β-CD, etc. [12,13]. Polymeric CD consists of native CD linked with one (dimeric) or more (polymeric) spacers. These structures allow improved drug-binding abilities [14]. The truncated cone shape of CD with an internal hydrophobic cavity and external hydrophilic surface [15] makes the formation of inclusion complexes with hydrophobic compounds possible, such as curcumin [4,6], budesonide [16], prilocaine [17], etc., increasing their water solubility and bioavailability. Besides, CD can also improve thermal and physical stability and reduce drug toxicity [15]. Despite these advantages, in some cases, the amount of CD that is required to dissolve the drug is high, and can trigger toxic effects [14].

To date, liposomes continue to be among the lipid nanocarriers that are most used for the drug delivery of hydrophilic and lipophilic compounds [18]. Among the different types of these lipid vesicles, those containing a single bilayer (unilamellar vesicles or ULV) are ideal to encapsulate hydrophilic drugs, while those containing multiple bilayers (multilamellar vesicles or MLV) are more suitable for lipophilic drugs [19]. 

Since 1994, drug-in-cyclodextrins-in-liposomes (DCL) has been employed as a drug delivery strategy to increase the loading capacity of hydrophobic drugs into the liposomes. The addition of the lipophilic molecule into the lipid bilayer and the inclusion complex drug CD into the aqueous compartment of liposome, by double-loading (DL), gives rise to DCL–DL. This last approach has several advantages such as, in particular, the prevention of the rapid clearance of the drug, when injected, due to the affinity of other blood components with CD [12]. In addition, the combination of both systems (CD and liposome) is able to improve the encapsulation efficiency of drugs [15]. The coupling of both delivery systems, by encapsulating a drug CD inclusion complex into liposomes, is proposed to circumvent the drawbacks of each separate system. A very interesting review about these nanoparticulate systems has been reported [15]. Our research group has wide experience in the use of this combined approach in order to increase the therapeutic efficacy of anesthetics and anti-inflammatory compounds [20,21,22]. Moreover, DCL–DL enhanced the amount of the drug in stratum corneum and deeper skin layers compared to conventional liposomes (CL). From the studies, we concluded that liposomes in the presence of CD may be a promising carrier for the effective cutaneous delivery of drugs [20,21]. In the last years, many studies have been published with the aim of improving the water solubility, chemical stability, and bioavailability of drugs [23]. This is the case of using this combination strategy for paclitaxel to improve loading into liposomes and pharmacokinetics due to the increased aqueous solubility of paclitaxel complexed in DMβCD as compared to pure drug solubility [24]. Wang et al. have used this strategy for increasing risperidone stability, achieving a second release phase of the combined delivery system that is slightly slower, after an initial burst release, and potentially suitable as a long-acting injection formulation [25].

DCL–DL is also being investigated for loading substances to be used as additives for providing the photoprotection and preservation of essential oils [26] or food nutraceuticals [27,28].

Although many studies are being realized with this combined approach, most of them have centered the research in how the presence of cyclodextrin affects the drug solubility, the physicochemical properties of resulted vesicles, the stability, and the in vitro release. Other authors make reference to the use of drug delivery systems combining the advantages of cyclodextrin inclusion complexes and those of deformable liposomes that are mainly targeted for skin delivery, with the purpose of increasing the diffusion of the encapsulated drug [29]. The enhanced therapeutic efficacy of drug-in-cyclodextrin-in deformable liposomes has been confirmed by our studies, achieving favorable effects of drug CD complexation and allowing a significant enhancement of intensity and duration of anesthetic effect with respect to those that are single-loaded [22].

Taking into account the scarce aqueous solubility of Cur, it is predictable that this molecule is placed on the lipid moiety of the liposome, specifically near the acyl region. In addition, it is known to interact with the bilayer, ordering or disordering it as a function of the presence of cholesterol [30]. Some years ago, Jaruga et al. [31] and Jaruga et al. [32] demonstrated that Cur highly impacts the permeability of membranes, in the sense of increasing it [31], and provoking changes in membrane fluidity [31,32]. However, only a few articles make reference to the effects of the presence of CD in Cur double-loaded into liposomes on the lipid bilayer, drug entrapment, and drug release. Therefore, in an attempt to minimize the effects of Cur on the bilayer, and at the same time, increase the drug entrapment efficiency, Cur-loaded DCL–DL were formulated. Firstly, we aimed to optimize the lipid bilayer composition of this combined formulation and secondly, the analysis of the effects of the CD and Cur over some bilayer properties, drug solubility, and drug stability, was carried out. Finally, the presence of CD into lipid vesicles on Cur release behavior was also investigated.

## 2. Materials and Methods

### 2.1. Materials

Sigma-Aldrich Co (Barcelone, Spain) supplied Curcumin (Cur), didodecyldimethylammonium bromide (DDAB), cholesterol (Cho), cholesteryl hemisuccinate (Chems), l-α-Phosphatidylcholine from egg yolk (EPC) and 1,2-Dipalmitoyl-*sn*-glycero-3-phosphocholine (DPPC). Hydroxypropyl)-β-cyclodextrin (HPβCD) was purchased from Roquette (Lestrem, France). Other chemicals were high-quality analytical. Solvents were HPLC quality.

### 2.2. Quantification of Curcumin

Cur concentration was measured by HPLC (Hitachi Elite LaChrom, San Jose, CA, USA). The analytical method was optimized by using a column Agilent Zorbax SB C-18 4.6 × 150 mm, 3.5 µm, by following a method previously proposed by Musfiroh et al. [33]. The mobile phase consisted of acetonitrile:acetic acid 2% (50:50 *v*/*v*). The flow rate was fixed at 1.2 mL/min, and had an injection volume of 20 µL. Absorbance was measured at 420 nm. With these conditions, peak areas were measured, and HPLC analysis was conducted at room temperature.

UV/VIS spectrophotometry was used in formulations containing HPβCD. Cur content was measured by using an Agilent 8453 UV-visible spectrophotometer (Agilent Technologies, Budapest, Hungary). A 200-µL sample were diluted up to 5 mL with an acetonitrile-acetic acid 2% 1:1 *v*/*v* mixture, and absorbance was measured at 425 nm.

### 2.3. Phase Solubility Studies

These studies were performed as Higuchi and Connors described [34]. This approach examines the effect of HPβCD (ligand) on the Cur being solubilized (substrate). Briefly, increasing concentrations of HPβCD (0 mM, 5 mM, 10 mM, 15 mM, 20 mM, and 25 mM) were dissolved in MilliQ water in closed vials. To each vial, 25 mg of Cur was added to reach saturation. Vials were shaken over seven days at room temperature and were protected from light. Samples were filtered through 0.45-µm nylon filters and measured by UV/VIS. All of the samples were performed in duplicate.

Theoretically, we assumed that one Cur molecule may form a complex with *n* HPβCD molecules; then, the complexation equilibrium is as follows:

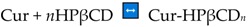

The equilibrium constant *K*_c_ (complexation constant) is expressed as follows: $$K_{c}=\frac{\left[Cur-HPBCD\right]}{\left[Cur\right]-{\left[HPBCD\right]}^{n}}$$
where [Cur] is the concentration of free drug in the solution and, at saturation, [Cur] is the solubility in water of the free drug *S*_0_:(1)$$\log\left(S_{t}-S_{0}\right)=n\log\left[\mathrm{HP}\beta \mathrm{CD}\right]+\log\left(K_{c}S_{0}\right).\;$$
where *S*_t_ is the measured solubility in the presence of cyclodextrin solutions.

The *n* term was obtained from the slope $\log\left(S_{\mathrm{tot}}-S_{0}\right)$ versus log[HPβCD]. A complexation constant was obtained from the intercept with the *y*-axis.

Once the drug:cyclodextrin stoichiometric ratio was determined, the apparent stability constant for cur–HPβCD complexes was obtained from the phase-solubility diagrams. In the case of 1:1 ratio (*A*_L_ profile), *K*_s_ was calculated as follows:(2)$$K_{1:1}=\frac{slope}{S_{0}\left(1-slope\right)}$$

Data are plotted as the mean ± SD of three independent experiments.

However, for the *A*_p_-type system and assuming that Cur-HPβCD complexes of 1:1 and 1:2 both coexisted, the stability constants for these two complexes can be estimated from the equation [35]:(3)$$\frac{S_{t}-S_{0}}{\left[HP\beta CD\right]}=K_{1:1}K_{1:2}S_{0}\left[HP\beta CD\right]+K_{1:1}S_{0}$$
where *K*_1:1_ and *K*_1:2_ are the stability constants of 1:1 and 1:2 inclusion complexes, which can be calculated from the slope and the intercept of the linear curve generated. 

The solubility enhancement ratio (ESR) was calculated from the following equation:$$ESR=\frac{Cur\;solubility\;with\;HP\beta CD}{Cur\;solubility\;without\;HP\beta CD\;}$$

Cholesterol (Cho) is one of the main components of liposomes, and it is known to bind well with cyclodextrin, displacing the drug from the hydrophobic core [12]. To further analyze this replacement, another phase solubility study was performed, adding to each vial 20 mg of Cho and equal molar concentration of Chems. Samples were treated as described previously. 

### 2.4. Preparation of Cur–HPβCD Complex

The complex was formed by dissolving 14.5 mM of HPβCD and 0.4 mM of Cur in citric acid–disodium phosphate buffer at pH 5.4 and stirring the solution over 72 h. This buffer was selected with the aim of maintaining Cur stability. Afterward, the solution was centrifuged at 1000 rpm 10 °C 10 min. The supernatant was collected, and its Cur content was measured.

### 2.5. Preparation of Liposomes

Liposomes were prepared by thin layer evaporation technique (TLE) as previously described [20]. In brief, lipid components (EPC, Cho, Cur, and SA or DDAB) were dissolved in a mixture of methanol (3.2 mL) and chloroform (4.8 mL) in a round-bottom flask. The organic solvent was evaporated under vacuum using a rotary evaporator (Büchi R-210 with Heating Bath Büchi B-491, Flawil, Switzerland) until a complete dried lipid film was formed. The fixed temperature in this process was 58 °C, which was above the transition temperature of EPC. Once achieved, three mL of buffer solution (Mellvaine’s citric acid/phosphate buffer pH 5.4) was added, and the film was hydrated until MLV formation. The formulation was rapidly sealed and stored in darkness at 4 °C. 

Drug-in-cyclodextrin liposomes by double-loading technique (DCL–DL) were prepared as above. In this case, cholesteryl hemisuccinate (Chems) was added instead of Cho as the lipid component at different molar ratios, as indicated the experimental design. Cur–HPβCD complex (0.04 mM Cur, 3 mL) was incorporated in the aqueous phase of vesicles. The formulation was rapidly sealed and stored in darkness at 4 °C.

As the first objective of this work was to evaluate the effect of the Cur–HPβCD complex on the lipid bilayer of the prepared liposomes, an experimental design was proposed for optimizing the lipid composition of the liposome bilayer in terms of its stability.

Four formulation parameters, named as factors (Chems concentration, DDAB concentration, Cur concentration, and phospholipid type) and three levels of each factor were introduced for generating a L_9_ Taguchi orthogonal array. Factors and their levels, as well as the experimental matrix, are shown in Table 1 and Table 2. The values of these variables were selected on the basis of our experience as well as from the literature. 

Once the experimental data were obtained, the ANOVA test was performed to determine which test parameters were statistically significant for every response variable. The analysis of mean (ANOM) was also realized in order to determine those factors that affect the response and compare the relative strength of the effects. 

The effect of each variable on the different responses was calculated by using the following equation:E(Xi)=∑Y(+)i−∑Y(−)iL2
where *E*_(*X_i_*)_ is the effect of levels of the tested variables, *Y*_(+)*i*_ and *Y*_(−)*i*_ are the response variables from the experimental runs in which the variables being tested are added at their maximum and minimum levels, respectively, and L is the number of experiments that are carried out. When the value of effect (*E*_(*X_i_*)_) of the tested variable is positive (>0), the influence of the variable is greater at the high level, while when it is negative (<0), the influence of the variable is greater at the low level [36].

The Pareto chart shows the absolute values of the standardized effects. On the basis of these charts and analysis of variance (ANOVA) results, we established the variables that exhibited significant main effects on the selected responses.

Finally, optimization techniques have been applied in order to obtain a predictable response as a function of the desired properties. In this work, minimizing the vesicle size and polydispersity index (PdI), and maximizing the PDE and zeta potential, were desirable. Regression analysis has been tested as a mathematical tool in this stage.

### 2.6. Liposome Characterization

#### 2.6.1. Liposome Size, Polydispersity Index, and Zeta Potential

Liposome size and PdI were determined by dynamic light scattering technique by using Zetasizer Nano-S equipment (Malvern Instruments, Malvern, UK). Results of size were expressed as the average liposomal hydrodynamic diameter (nm). Values of the dimensionless parameter PdI less than 0.5 indicated a homogeneous and monodisperse population.

The surface charge of vesicles was determined by correlation spectroscopy from electrophoretic mobility (*μ*) measurements, by using the same equipment as above. Results were expressed as zeta potential (*Z*, mV) after conversion of *μ* to *Z* by the Smoluchowski equation: *Z* = *μη*/*ε*, where *η* is the viscosity, and *ε* is the permittivity of the solution.

Both measurements were made at room temperature, and 200 µL of samples were diluted with citric-phosphate buffer solution (1/20).

#### 2.6.2. Encapsulation Efficacy

In order to determine the amount of Cur entrapped, encapsulation efficacy (EE) was measured. An aliquot of sample was centrifuged at 10000 rpm 4 °C for one hour. Afterwards, the resulting pellet was treated with sodium lauryl sulfate followed by three cycles of sonication (10 min) and vortex (one min) to disrupt the vesicles. Absorbance of the samples was obtained as mentioned in Section 2.1.

### 2.7. Antioxidant Activity

The aim of this assay was to measure the antioxidant activity of Cur due to its ability to scavenge free radicals generated in aqueous and lipophilic phases [37]. ABTS (2,2′-azino-bis(3-ethylbenzothiazoline-6-sulfonic acid) was used as the oxidizing agent in an electron transfer-based reaction, as was reported by Pisoschi and Negulescu [38]. This oxidizing radical (ABTS^•+^) was obtained after an electron from the nitrogen atom of ABTS was lost. For this, 2.98 mM of ABTS and 0.98 mM of K_2_S_2_O_8_ (potassium persulfate) in purified water were mixed. This solution, which has a dark blue color, is reduced by an antioxidant into colorless ABTS because the nitrogen atom quenches the hydrogen atom, yielding the solution decolorization. The decrease in the absorbance values was monitored, and Trolox (1.05 mg/mL in ethanol absolute) was chosen as the standard antioxidant.

The antioxidant activity of Cur was calculated to the total amount or Cur added to the sample after destroying it with sodium lauryl sulfate, and to the supernatant that resulted after the centrifugation process.

Afterwards, working solutions were prepared into 96-well plates, adding serial dilutions of each sample and placing 10 µL of each of them with 90 μL of ethanol to react with the fresh ABTS solution (100 μL). Trolox standard stock solution was also prepared in ethanol and was equally diluted. Then, the absorbance at 734 nm was measured six minutes after initial mixing (Synergy HT Plate Reader, Winooski, VT, USA). All of the measurements were performed in triplicate. 

The antioxidant activity was defined as EC50, which was indicative of the equivalent concentration of antioxidant to decrease the initial concentration of ABTS in 50%. Finally, this parameter was expressed as EC50 compound/EC50 Trolox [39].

### 2.8. Differential Scanning Calorimetry (DSC) and Hot Stage Microscopy

Thermal analysis was performed using a DSC (DSC Q20, TA Instruments, New Castle, DE, USA). Samples for DSC were prepared from pure components of the formulation and physical mixtures of all of them with Cur and liposomes. Samples were placed in TA Instruments standard aluminum crucibles with lids of 20-µL capacity. Next, the vessels were tightly closed using a special press. The reference crucible was air in all of the samples except for liposomal formulations, in which the vessel was filled with buffer solution. The scanning was conducted between 25–200 °C at a heating rate of 10 °C/min. The thermal behavior of samples was evaluated by determining the melting temperature (°C), and by calculating the enthalpy (kJ/mol). Thermograms of heat flow versus temperature were depicted.

In addition, different observations were made during heating using a hot stage microscope (HSM Mettler model FP82HT, Greifensee, Switzerland). For this assay, about 10 mg of samples (raw materials and physical mixtures) were placed on glass slides with coverglass and heated at 5 °C/min.

### 2.9. Transmission Electron Microscopy

Morphological studies were carried out by transmission electron microscopy (TEM) (ZEISS LIBRA 120, Oberkoche, Germany). All samples (10 μL) were previously diluted with one mL of citric/phosphate pH 5.4 solution. Then, a drop of the diluted sample was left to dry on a microscopic copper-coated grid (transmission electron microscopy grid support films of 300-mesh Cu). After drying completely, a drop of an aqueous solution of uranyl acetate (1% *w*/*v*) was added for negative staining. Eight minutes later, the excess solution was wiped with filter paper and washed with purified water. Then, the specimen was viewed under the microscope with an accelerating voltage of 75 kV at different magnifications.

### 2.10. Deformability

The bilayer elasticity of the prepared vesicles was measured by the extrusion method as reported earlier [40]. Briefly, samples were extruded two times through 800-nm pore sized polycarbonate membrane filters equipped in a Lipex Thermobarrel extruder (Northern Lipids Inc., Burnaby, BC, Canada) under air flow. After samples were homogenized, the recovered volume was measured. The deformability index (DI), which is also named elasticity, was calculated as a relationship between the flux of the sample and the ratio of the measured size and the pore size. To obtain the flux (*J*_flux_), the recovered liposomes from previous extrusion was extruded through a 100-nm pore size polycarbonate filter (*r*_p_) by applying a pressure of 40 bar for five minutes. The averaged liposome diameter (*r*_v_) was measured by DLS. 

The elasticity of the vesicles was calculated from the following equation:$$DI=J_{flux}{\left(\frac{r_{v}}{r_{p}}\right)}^{2}$$
where *J*_flux_ is the rate of penetration through a membrane filter (the volume of sample extruded in one minute); *r*_v_ is the vesicle size (after extrusion); and *r*_p_ is the size of the membrane pore.

### 2.11. Phosphorus Content

To further analyze the integrity of the bilayer, the phospholipid recovery after extrusion was also measured. In this case, a Stewart assay was performed to determine the DPPC content before and after extrusion.

Stewart assay is based on the ability of phospholipids to form a complex with ammonium ferrothiocyanate. As an advantage, this protocol avoids interferences with inorganic phosphates, which allows samples to be suspended in any buffer containing phosphate salts [40]. The protocol was previously described by Zuidam et al. In brief, a standard solution (0.1 M) of ammonium ferrothiocyanate was prepared by adding 27.03 g of ferric chloride and 30.4 g of ammonium thiocyanate into one L of deionized distilled water. At the same time, a standard solution of DPPC (0.1 mg/mL) was prepared in chloroform. For the calibration curve, increasing amounts of lipid standard was added to chloroform up to 2 mL, and the same volume of ferrothiocyanate standard. After vortexing for 15 seconds, samples were centrifuged for five minutes at 1000 rpm and spectrophotometrically quantified at 465 nm. For the liposomes samples, the same procedure was performed before and after extrusion [41].

### 2.12. In Vitro Release Studies

In vitro release tests of Cur from liposomes were carried out by the dialysis method. In this study, 1.5 mL of liposome dispersion was placed in a dialysis bag (Spectra/Por 4, molecular cut-off 12–14 kD, Califormina, CA, USA), which was previously rinsed and soaked for 30 min, and both borders were sealed with a dialysis clip. The device was then incubated in 500 mL of dialysis solution. This solution was composed by 25% ethanol and 0.5% Tween^®^-80 (*v*/*v*) in a citric/phosphate buffer pH 5.4. The dialysis bags were stirred at 200 rpm at 25 °C (IKA^®^ RT10). At predetermined intervals of 0.5 h, 1 h, 1.5 h, 2 h, 3 h, 4 h, 5 h, 6 h, and 24 h, 50 µL was taken for the dialysis bag and diluted with acetonitrile:acetic acid 2% 1:1 (*v*/*v*). Quantification was performed with a spectrophotometer at 425 nm. The assay was performed in the absence of light. 

The results were expressed as the percentage of Cur released from liposomes using the following equation:$$\%\;Cur\;released=\frac{Cur_{i}-Cur_{t}}{Cur_{i}}\times 100$$
where Cur_i_ is the initial concentration of Cur in the dialysis bag, and Cur_t_ is the concentration of Cur in the dialysis bag at time t.

From the experimental data, three model-dependent approaches were used to compare the Cur dissolution profiles. The model-dependent approaches included the first-order, the Higuchi, and the Korsmeyer-Peppas models.

In addition, model-independent methods were applied in order to generate a single value from a dissolution profile, thus allowing data to be compared directly. In this study, for each sample, the percentage dissolution efficiency (DE%) is calculated as the percentage ratio of the area under the dissolution curve up to time t (AUDC_0_^t^), and that of the area of the rectangle described by 100% dissolution at the same time point (Q_100·t_), and is defined as follows:$$DE\%=\frac{AUDC_{0}^{t}}{Q_{100\cdot t}}\cdot 100$$

## 3. Results

### 3.1. Cur into Cationic Lipid Bilayer

#### 3.1.1. Compatibility of Curcumin-Charged Lipids

In this study, the development of cationic liposomes was proposed. As it is well-known, SA and DDAB can provide cationic charge to the vesicle surface, avoiding aggregation phenomena. Cur was added into the lipid bilayer in all of these formulations. The aim of this first section was to provide additional information about the drug behavior into the bilayer in the presence of these cationic agents, in terms of compatibility and stability.

A first significant difference was appreciated in the lipid film color due to the presence of SA or DDAB (Figure 1a). SA samples showed a red color when methanol was added, and this mixture returned to a yellow color again when chloroform was incorporated; once the solvents were removed by rotaevaporation, the film returned to a red color. On the contrary, samples with DDAB exhibited a yellow color during the whole process, and any change was detected with the addition of organic solvents. Similar behavior was observed with Cur sample, in the absence of a charged lipid.

The difference in the polarity of both media can affect the Cur ionization, and then, the drug color. It is well-known that Cur is very slightly soluble in water, and solubility is pH-dependent. In the keto-enolic balance, the enol form has the ability to lose one proton at pH values between 7.5–8.5 (Figure 1b), and a change in color from yellow to orange appears when the pH is higher than these values [42]. The relevance of this visual result on the compatibility and stability of Cur into the lipid bilayer was analyzed by several techniques. 

Thermal analysis provides information about the compatibility between Cur and the charged lipid, whether SA or DDAB. This study was performed using a DSC analyzer, and the obtained thermal curves are shown in Figure 2.

The DSC thermogram of pure Cur showed an endothermic peak at 178 °C corresponding to its melting point. According to some authors, the shape of this profile corresponds to Form I, which is the most usual one among the different polymorphic forms of Cur [43]. After dissolving Cur in methanol:chloroform and evaporating the solvent under vacuum in the rotary evaporator, samples showed that although the endothermic peak appears at 176 °C, a new exothermic peak also appeared at 114 °C. This sequence of peaks (exothermic peak followed by endothermic one) is typically observed upon recrystallization of the product. Therefore, the appearance of this phenomenon would indicate that after solvent evaporation, Cur precipitates in an amorphous form [44] which, upon heating, recrystallizes to Form I, which is the most stable form. DSC diagrams corresponding to the physical mixture Cur+SA and Cur+DDAB showed that in the solid dispersion that is made, only the endothermic effect associated with the melting of the carrier was detected. Since no additional endothermal effects appeared and considering that commercial Cur is a crystalline powder, the absence of its typical endothermal peak of fusion into the DSC thermogram was interpreted as a “solid solution” formation of drug into the vehicle.

The complementary study of these systems by HSM showed that after dynamic heating at 90 °C, the carrier melts, and small and larger vesicles are observed. This dissolution process is broader as temperature increases, as observed by HSM producing a single liquid phase enriched in Cur, confirming DCS findings. Results evidenced that a solid solution was obtained because the drug was solubilized in the carrier. The dissolution process of Cur in both cationic agents provides different solutions in colors, being redness in the case of SA and yellow in the DDAB sample (Figure 2c,d), respectively).

#### 3.1.2. Bilayer Stability

In a first study, a prospective analysis of the Cur stability into the lipid bilayer and the effect of the cationic agent were studied. The physicochemical parameters that were evaluated included the vesicle size, polydispersity index, zeta potential, and encapsulation efficiency. Also, the maintenance of the antioxidant activity of Cur was evaluated.

Physicochemical properties of liposomal formulations are depicted in Figure 3. The study was carried out for four weeks. Concerning these properties, a decrease in the surface charge over time was evidenced in formulations with charge. However, neutral formulations were characterized by the constant values of this parameter during the four-week study, according to the bibliography. The addition of lipid compounds into the bilayer causes an inner rearrangement of lipids, giving rise to a decrease in the zeta potential [45]. 

In addition, the neutralizing effect of Cur at pH 6.5, which is partially ionized at this pH, and the formation of aggregated and fused structures with time may contribute to these reduced zeta potential values, which can be reverted in a sign in the case of SA. This hypothesis was corroborated from the size distribution (Figure 3b), which acquired more dimensions as the time increased. Although there are no significant differences, it is possible to appreciate higher sizes in the case of SA vesicles, which is probably attributable to lower stability of lipid bilayer with this lipid, resulting in higher PdI values (Figure 3d). However, in DDAB formulations, despite the size increase over time, the PdI did not increase enough (from 0.2 to 0.4), and a more stable lipid bilayer was generated with this lipid [46].

All of these effects contributed to a decrease of Cur entrapped over time, and this tendency was more pronounced in SA samples. It is important to note that in charged formulations, an increase of the drug that was entrapped was obtained after the third week. This capability of entrapping Cur can be related to the ionized molecule, the vicinity to the vesicle surface, and the affinity of the drug for nonpolar media. This behavior has been yet described with fluoroquinolones [47]. In addition, the surfactant properties of DDAB could enable the inclusion of the drug into the bilayer.

The antioxidant activity has been expressed as EC50, which is defined as the equivalent concentration of antioxidant that is capable of decreasing the initial concentration of ABTS in 50%. This parameter was referred as “equivalents in Trolox”, which is a water-soluble carotene derivative that was used as the reference substance. This antioxidant activity was calculated to the Cur entrapped after disrupting liposomes with sodium lauryl sulfate, and to the supernatant after extracting the Cur-loaded liposomes. Regarding the percentage of Cur entrapped in each batch, we pointed out this result to predict the existence of free Cur in the aqueous medium and Cur entrapped into the vesicles.

The results of this assay are presented in Figure 4. Firstly, we can observe higher values of antioxidant activity than Trolox (corresponding to values lower than one) in almost all of the formulations, which indicated the maintenance of the drug activity over time. This activity was reinforced by the preservation of the free drug in the aqueous medium. Certainly, a stabilizing effect of liposome on Cur was shown, because the antioxidant activity in these formulations was maintained over time, in spite of presence or absence of surface charge, compared to the Cur solution, which exhibited higher EC50 values.

On the other hand, a lack of Cur antioxidant activity in the supernatant medium was obtained in all of the samples, except in DDAB formulations, which indicated the loss of antioxidant activity in those formulations without charged lipids and with SA. However, DDAB formulations maintained a significant antioxidant activity on the ABTS (Figure 4a). 

As a conclusion, for the next stages, DDAB was selected for joint formulation with Cur. The higher stability of the vesicle sample within one month (Figure 4b), as well as the maintenance of the antioxidant activity of Cur, makes this lipid surfactant suitable for developing double-loaded liposomes. 

### 3.2. Cur in HPβCD-in Double-Loaded Liposomes (DC–-DL)

Once DDAB was selected as the cationic-charged lipid into the lipid bilayer, we proceeded to study the inclusion of Cur into the aqueous phase of liposomes. Previous studies suggested the use of CD for this approach, specifically HPβCD. Then, first of all, a preliminary phase solubility diagram was designed in order to predict the CD behavior in terms of the capture of lipid compounds from the vesicle bilayer. Afterwards, the optimization of the lipid bilayer was carried out, and finally, a complete physicochemical characterization of the optimized formulation was performed in order to analyze the influence of CD on these final properties. 

#### Phase Solubility Diagram Cur–HPβCD

The phase solubility profiles are shown in Figure 5. At the concentrations that were used for this study, a positive deviation from linearity is evident, suggesting the formation of a higher-order complex with respect to HPβCD (i.e., one Cur binds to more than one HPβCD). 

Equation (1) indicates that the plot of log(*S*_tot_-*S*_0_) versus log [CD] should be a linear curve with a slope close to *n*. Estimation via this approach yielded a *n* value of approximately 3.72 (*R*^2^ = 0.953), which is higher than 1, indicating that higher-order complex(es) are formed in the binary system Cur–HPβCD. In addition, the coexistence of Cho and the binary mixture generated a decrease in the Cur solubility as a consequence of a displacement of the drug from the cyclodextrin cavity by the Cho, as has been reported by several authors [16,48,49]. For this reason, Cho was replaced in the liposome composition by Chems, which is a steroid derivative. In this case, the presence of Chems besides Cur and HPβCD leads to increased Cur solubility. The lower *n* value, which is next to two, together with the surfactant nature of Chems, contributes to the higher value of Cur solubility. 

From Equations (2) and (3), apparent stability constants (Table 3) showed higher values for Cur + Chems in the presence of CD. Chems also significantly contributes to enhance Cur solubility up to a ratio next to 700, compared to the drug without cyclodextrin at 25 mM of HPβCD. On the other hand, Cho very significantly reduced Cur solubility at this CD concentration. These results have been depicted in Figure 6. 

On the basis of these advantageous results of adding Chems to the lipid bilayer instead Cho, this steroid derivative was selected for the subsequent studies.

### 3.3. Optimization Stage of Lipid Bilayer Composition

The Taguchi orthogonal model was selected to screen the effects of Chems, DDAB, Cur, and phospholipid, which are all lipid bilayer constituents, on the size distribution and zeta potential of liposomes. This experimental array generated a worksheet with nine formulations (Table 2). 

The physical characteristics of the liposomes, i.e., size, PdI, and zeta potential, were determined (Table 2). The samples showed a high variability in all of the responses. Sizes ranged from 1039 nm to 6229 nm; PdI varied from 0.46 to 1; and zeta potential values ranged from −3.7 mV to 23 mV. 

In order to investigate the statistical significance of these factors on the evaluated responses, experimental data were analyzed from the ANOM of the main factors for each response (data not shown). After selecting those variables with statistical significance from ANOVA (Table 4), the contribution analysis of them was explored from the Pareto diagrams (Figure 7). The use of these charts is aimed to determine the magnitude and the sign of the effects. The critical values indicating the statistically significant effect of factors at a 99.5% confidence level were 732 and 1.963 for the vesicle size and zeta potential, respectively. Effects above this critical limit are significant, and effects below this value are not likely to be significant.

Concerning the vesicle size (Figure 7a), a clear effect of the phospholipid was obtained (X6). As the effect was negative and as minimizing the response was desirable, the middle level (0) was selected. In addition, Cur loaded into the lipid bilayer had a great influence on the vesicle size in terms of reducing it when one mg or three mg were used (X8). Both values of drug amounts were also desirable for increasing the zeta potential of liposomes, as shown in Figure 7b. Similarly, low level of Chems must be fixed in order to optimize this parameter (see X1 and X5 in Figure 7b). Finally, the positive effect of DDAB (X2) allowed selecting 0.021 mmol for the following studies.

From these charts, we can predict the suitable conditions and formulation composition that minimize the vesicle size and PdI, and maximize the zeta potential. On the basis on these determinant facts, levels of factors were fixed as follows: Chems 0.0127 mmol, DDAB 0.021 mmol, Cur 1 mg, and DPPC as the phospholipid.

The use of regression analysis as the optimizing tool enables the study of the influence of the main factors and their interactions on the response evaluated. As was previously reported, from the regression equations and model fitting parameters, the optimized parameters and predicted responses were obtained; the data are shown in Table 5. Afterwards, a confirmatory experiment was performed for each response, and the prediction error was calculated. 

From the equations, it is important to note the bad *R*^2^ adjust (0.006) that was obtained in size compared to the zeta potential (0.8452). The accurate checking of these predictions was carried out after preparing the optimized formulations. As shown in Table 6, experimental results were comparable to the predicted ones. However, the adjustment gives rise to less prediction error in the case of zeta potential when confirmatory experiments were developed. So, the regression analysis provides an adequate optimizing technique with lower error prediction.

The response surface plot for zeta showed the influence of Chems and DDAB factors when Cur was maintained fixed in one mg (Figure 8). The maintenance of this value for Cur was justified by the low contribution of this factor on the response that was evaluated. From these plots, a clear decrease in the zeta potential of the vesicles was evident as the Chems content increased. However, and taking into account that DDAB provides a significant cationic charge density to the structure, the increase in zeta potential was slightly modified by this factor. 

Finally, the percentage of Cur entrapment in the optimal DCL–DL formulation was 52.38%. Taking into account that the percentage of Cur entrapped into the lipid bilayer in the absence and in the presence of HPβCD in the aqueous medium was 92.27% and 84.04%, respectively, a clear influence of the Cur–HPβCD complex into the aqueous phase was evidenced. However, as will be discussed in the next section, despite this lower entrapment of the drug, the existence of combined nanostructures in the formulation could provide advantages in terms of deformability and Cur release.

### 3.4. Characterization Studies of Cur-in-HPβCD in Doubled-Loaded Liposomes

#### 3.4.1. Morphological Analysis

TEM images of Cur-loaded liposomes in the bilayer are named conventional liposomes (CL), CL containing HPβCD into aqueous phase (CL-HPβCD), and DCL-DL, and are respectively presented in Figure 9a–d. TEM images revealed their nanometric sizes. Conventional liposomes without HPβCD are spherically shaped (Figure 9a), and the samples showed crystallization features, which were probably of Cur (Figure 9b). Besides, the TEM images of formulations containing HPβCD showed that the spherical shape was not preserved, and deformable vesicles were obtained. 

On the other hand, the results displayed that small spherical nanostructures appeared in the presence of cyclodextrin (Figure 9c,d). These structures may be the consequence of the interaction of HPβCD with different lipid molecules of the bilayer, which are displaced outside of the vesicle. 

#### 3.4.2. Effect of HPβCD on Vesicle Deformability

From TEM analysis, it is evident that the ability of liposomes to maintain their spherical shape had to be questioned. As cyclodextrins are known for provoking changes in the integrity of the bilayer [50], their impact onto the flexibility of the optimized bilayer composition was analyzed. 

Three control batches were prepared, one in absence of HPβCD and Cur (empty), one containing Cur in the lipid compartment and buffer as aqueous media (CL), and another containing Cur in the bilayer and a 14.5-mM buffer solution of HPβCD (CL–HPβCD). The optimal formulation was performed in triplicate (DCL–DL). Each batch was extruded two times through an 800-nm pore size filter to homogenize the dispersion. Afterwards, the sample recovered was extruded once again through a 100-nm pore size filter. The recovered volume was measured. 

Figure 9e depicts the influence of HPβCD over the flexibility of the bilayer. Empty liposomes exhibited a DI close to one, which was higher than the DI of ultradeformable liposomes formulation that was proposed by González-Rodríguez et al. [40]. When Cur was added to the bilayer in the same concentration as was selected for the optimal formulation, the flexibility of the bilayer reduced in half compared to the previous batch. As the presence of Cur is the only difference between the two batches, it can confirm that this drug induces a very significant increase on the stiffness of the bilayer, which is also in accordance with the bibliography [30]. The addition of HPβCD to the system (CL–HPβCD) provokes an extremely significant increase in the flexibility of the bilayer, which can be related to the extraction of Cur and other components from the bilayer. However, when HPβCD is complexed with Cur (DCL–DL), its capability to hijack components of the bilayer was highly diminished, resulting in a flexibility close to the flexibility of the empty batch. 

To complement this study, the phosphorus content of liposomes after extrusion was also measured. In all of the cases, the recovery was close to 100%, which indicated that any liposomes were broken during the extrusion process, confirming the results obtained earlier. 

#### 3.4.3. Effect of HPβCD on Cur Antioxidant Activity

Cur is known for its capability to prevent the oxidation of other components. To measure its performance over time, an ABTS experiment was performed. This procedure compares the antioxidant capability of any substance with that of Trolox, and is expressed as EC50. Data should be interpreted as the smaller the EC50, the higher the antioxidant capability of the substance. In all of the cases, the EC50 of Trolox equals one. 

This assay was performed over the same samples previously prepared over a one-week interval. Empty liposomes were also measured for their antioxidant capability in order to confirm that neither the buffer nor the lipid components of the bilayer exhibited antioxidant activity. As Figure 10 shows, at time 0, the EC50 values of liposomes with Cur into the bilayer (CL) was lower than the EC50 values of both the optimal formulation (DCL–DL) and the liposomes containing Cur into the bilayer and the HPβCD solution as aqueous medium (CL–HPβCD), indicating an augmented antioxidant capability of the former. However, these differences did not show statistical significance.

After one week, the EC50 values of the CL–HPβCD and DCL–DL samples diminished, while that of CL increased. When these data were compared with those obtained at time 0, a significant improvement on the antioxidant activity of Cur in DCL–DL and CL–HPβCD was observed. CL liposomes exhibited a reduction in the antioxidant capability of Cur, although it was not significant. This phenomenon can be attributed to the stabilizing effect of HPβCD. 

#### 3.4.4. Influence of HPβCD on In Vitro Drug Release

Lipid composition of liposome bilayer is known to be a key factor in its permeability. When loaded, this parameter is directly related with drug release [51]. As it has been demonstrated that HPβCD is capable of extracting components from bilayers, its effect on Cur release was studied. 

Figure 11 exhibits the Cur release profiles of different samples. The assay was performed with two different control samples. The first one consisted on an ethanolic solution of Cur with the same concentration as the liposome formula. The second control sample consisted of the HPβCD–Cur solution that was used as aqueous media for the liposomes. Control samples and liposomes formulations were performed in triplicate. Test conditions were chosen in order to respect sink conditions in the receiver compartment.

The release results showed that the ethanolic solution of Cur and the HPβCD–Cur solution behaved similarly, with 75% and 69% drug released, respectively. After 24 h, the drug was completely released from both solutions. Liposome formulations behaved differently from solutions. CL, CL–HPβCD, and DCL–DL showed an initial burst of 59%, 66%, and 50%, respectively. However, DCL–DL and CL exhibited a greater sustained release after two hours compared with CL–HPβCD. Nevertheless, after 24 h, DCL–DL released 90% of its content, which is significantly higher (*p* = 0.0351) than the amount of Cur released from CL (less than 80%). From these results, the sequence of formulation capability of controlling drug release was CL > DCL-DL > CL-HPβCD > complex solution ≈ ethanolic solution.

Comparing these profiles, a relationship between them and DI can be suspected. To confirm this hypothesis, a Pearson product moment correlation analysis was conducted comparing the area under the dissolution curve (AUDC) with DI. The correlation coefficient (*r*) that was obtained was 0.909, which had very statistical significance (*p* = 0.0045).

To further analyze the release assay, a dissolution efficiency (DE) test was performed at six hours and 24 h. As Table 7 shows, no significant differences between parameters were present except for the DE at 6 h of CL–HPβCD when compared to DCL–DL (*p* = 0.0489) and the DE at 24 h of CL when compared to CL–HPβCD (*p* = 0.0079).

In order to elucidate which kinetic model better suited the release profiles, three different models were selected: the Korsmeyer-Peppas model, the Higuchi model, and a first-order kinetic model. In order to differentiate which model provides a better fitting, the log of Cur release versus the log time plot, Cur release versus the square root of time plot, and the log Cur release versus the time plot were performed (data not shown), and their correlation coefficient (*R*^2^) values were analyzed. As can be deduced from Table 8, only the Korsmeyer–Peppas model provided an accurate fitting for all of the release profiles. 

The Korsmeyer-Peppas model describes drug release from polymeric systems [52], although it has been successfully used for predicting drug release from liposomes [51]. This model is represented by $Q_{t}=K\times t^{\eta }$ where *Q*_t_ is the fraction of drug released at time *t*, K is the release rate constant, and *η* is the diffusion exponent. The value of this exponent predicts the release mechanism, so values below 0.45 indicate a Fickian diffusion mechanism, while values between 0.45–0.89 indicate non-Fickian transport, *η* = 0.89 indicates Case II transport, and values higher than 0.89 indicate super Case II transport. To calculate this parameter, only the initial 60% of the drug release profile should be used. As Table 8 shows, the diffusion exponent is below 0.45 in all of the cases, indicating a Fickian diffusion mechanism of release. 

Comparing the release constant (K) from both solutions, it can be affirmed that the presence of CD decreased K, allowing Cur to be released from the solution slightly slower when CD is present. Liposomes formulations exhibited a different behavior when compared with solutions. CL and CL–HPβCD exhibited similar K values, indicating that even though the presence of HPβCD affected this parameter, this influence was not present on the liposomes. However, DCL–DL obtained the lowest K, indicating that the presence of Cur in both compartments of the liposomes diminished the amount of Cur that was able to be released. These findings are supported by the drug release profiles shown in Figure 11.

## 4. Discussion

In the present work, an attempt has been made to encapsulate the polyphenol Cur-in-HPβCD-in liposomes as a double-loaded system (DCL–DL). This strategy allows Cur to be loaded into both the phospholipid bilayer (free Cur) and the aqueous core of liposomes (Cur–HPβCD). These supramolecular lipid aggregates were investigated for determining the effect of CD in the bilayer containing Cur. 

Among the different charged agents for cationic liposomes, SA and DDAB are widely studied [53,54]. As a weak base, with three pKa (7.7–8.5, 9.5–10.7, and 8.5–10.4), Cur may be affected by those lipids, since they possess amino moieties that can affect the pH of dispersion. As Cur can change its color with the loss of the first proton [42], the interaction between those lipids and Cur was studied. These changes in solution were maintained after the evaporation of solvents during the initial step of liposome production. With the aim of discarding a feasible interaction between lipids and the drug, DSC and HSM studies were performed. From the results obtained, we can conclude that Cur got dissolved in the lipid vehicle during the formation of the lipid film, which can improve the assembly of the drug in the bilayer. This behavior might imply an increased stability. 

Characterization studies of these formulations enabled select DDAB as a cationic lipid agent. This lipid surfactant is widely used in the study of vesicles and other biomembrane models [55,56,57]. The conical shape that this lipid achieved in aqueous solution provided an increased stability of liposomes, and has also been extensively reported [58,59].

Once the Cur behavior in this lipid bilayer was analyzed, the next step was the study of the influence of the HPβCD–Cur complex over a bilayer already containing Cur. A lot of research has been made about the influence of CD on the vesicle bilayer properties [12,49,50]. However, no researchers are investigating the effect of the Cur–HPβCD complex on Cur-loaded liposomes.

The DCL–DL systems are being used by numerous authors with the aim of increasing the loading capacity of poorly-soluble drugs [20,21,22,24]. However, one of their possible drawbacks is that the presence of cyclodextrins in these lipid systems can give rise to the removal of some of lipid components due to the affinity of CD for lipophilic compounds [12]. In particular, many studies have evidenced the kidnapping of Cho by these oligosaccharide structures [49,60]. We investigated this issue, and a displacement of Cur from the CD cavity by Cho was actually found. In order to prevent any destabilizing effect on the lipid bilayer, we decided to substitute this steroid by a derivative that is widely used in liposome formulations, Chems. 

This hemisuccinate derivative is more water-soluble than Cho. Although this steroid has a similar chemical structure to Cho, the loss of one proton from the hemisuccinate moiety gives rise to a lower capacity of packing the lipid bilayer [61]. A phase solubility diagram containing Chems demonstrated that not only did Chems have a different performance than Cho [62], it also augmented *K*_1:2_ and Cur solubility. Thus, Chems was selected as a replacement for Cho.

EPC is commonly used as the main lipid in liposome formulation. This phospholipid is known to be prone to oxidation. As Cur is a potent antioxidant agent, it is not desirable to have any component that is sensitive to oxidation in formulation. For this reason, DPPC was selected as the synthetic saturated phospholipid. However, with the aim of evaluating the beneficial selection of this phospholipid compared with EPC, an additional study based on design of experiments was performed, in which the amounts of Chems, DDAB, and Cur were also included.

In order to optimize the formulation, three levels were fixed for each variable. Size, PdI, and zeta potential were evaluated. The results of this study demonstrated that Chems significantly affected the surface charge of vesicles, decreasing it as Chems concentration increased. As was previously mentioned, Chems is a succinic derivative. This provides the molecule with a pKa of 5.5. Some studies have already demonstrated that Chems’ capability of mimicking Cho behavior depends on the ionization of the structure [61,62]. In our case, the pH was adjusted to 5.4, which triggered one half of the ratio of Chems to be ionized, decreasing the surface charge of the liposomes. Another factor affecting the surface charge was the concentration of DDAB, as previously reported by González-Rodríguez et al. [54].

On the other hand, the vesicle size was affected by the Cur amount on the bilayer and the phospholipid that was employed. Nonetheless, the size alteration may be due to the contribution of other components in the bilayer, as reported in the regression equation (*r*^2^ adj 0.006). Moreover, TEM images of samples containing CD showed nanosized structures that were different from the liposomes, which could affect the values of the size distribution. This can explain the variability in sizes obtained from DLS analysis and TEM images. 

In an attempt to elucidate the reason behind the presence of these nanostructures visualized by TEM, three different samples were prepared, which all contained the same bilayer composition, but had differing aqueous phases: there was one formulation containing only the buffer (CL), a formulation containing the buffer and 14.5 mM of HPβCD (CL–HPβCD) and a formulation that was made after the optimal formulation deduced from the experimental design (DCL–DL). The images clarify that these structures only appear in the presence of CD, independently of the presence of Cur as a complex with CD. Also, images from CL–HPβCD and DCL–DL reported the non-spherical shape of liposomes, which can indicate the presence of deformable liposomes [63].

To demonstrate this hypothesis, the deformability of liposomes was tested by extrusion, which is a methodology that was previously described by González-Rodríguez et al. [40]. As mentioned before, empty liposomes possessed a DI close to one mL/min, indicating that the liposomes were extremely flexible. This characteristic was provided by Chems, which is able to increase the elasticity in membranes at a pH above pKa [61]. The presence of HPBCD solution in the aqueous compartment of the liposome increased the DI 2.5 times compared with empty liposomes. As described before, CD is capable of extracting lipid components to a degree in which holes in the bilayer are created [49,64]. It is not clear which component of the bilayer was extracting HPBCD, but it can be concluded from the phase solubility diagram that Chems was not removed due to its lack of interaction with this CD. Regarding DCL–DL, the deformability index was close to that of the empty liposomes, which leads to the conclusion that even though HPBCD increased the DI in liposomes, the Cur condensing effect [30,65] counteracted the CD effect. These results are in agreement with those in the literature [63].

Another positive effect of CD over the formulation was the maintenance of the the antioxidant capability of Cur. For this assay, antioxidant capability via an ABTS test was performed over freshly prepared samples and after one week. As can be observed in Figure 10, liposome formulations exhibited similar antioxidant capability at time 0, making it clear that HPβCD had no effect over this parameter. After one week, the CL formula maintained Cur antioxidant capability, which is in accordance with the literature [66]; however, CL–HPβCD and DCL–DL ameliorated the antioxidant activity of Cur, which indicates that a stabilization of Cur was occurring with the addition of this CD. This positive effect of CD over the antioxidant capability of the drug was previously reported by Malakzadeh and Alizadeh [67]. In this study, an amlodipine besylate-γ-CD complex showed an increased capability of scavenging free radicals due to an enfeeblement of the H–O covalent bond in the –OH radical of amlodipine as a consequence of the strong binding between the drug and CD [67]. In the case of Cur, a 1:2 Cur/CD ratio was observed; this implies that both ends of the molecule are entrapped inside the CD hydrophobic core, leaving the keto-enolic moiety exposed to the media [4]. Some authors have proposed that this keto–enolic moiety is attacked by free radicals during oxidation processes [42]. In addition, Cur–HPβCD binding was found to be really strong when Chems was present in the formulation, as previously reported in Table 3. Thus, it is possible that the Cur–HPβCD association is weakening the O–H bond, which will trigger an increase in the antioxidant activity of Cur. 

Regarding drug release, the liposome samples in all cases showed a biphasic release profile characterized by a burst initial release followed by a much slower release, which was probably due to drug diffusion through the bilayer. This performance was also described by Wang et al. (2016) who compared liposomes containing a risperidone–HPβCD complex with conventional liposomes containing the same drug [25]. As Figure 11 depicts, DCL–DL and CL samples performed a slower release when compared with ethanolic and HPβCD–Cur complex solutions. On the other hand, CL–HPβCD did not control drug release as much as DCL–DL and CL. 

Release behavior, in terms of magnitude of Cur released over time, was quantified by AUDC_0_^t^. This parameter was directly correlated with the deformability index of liposomes. In that way, as *r* is a positive value, an increase in the deformability of the liposome allows Cur to go through the bilayer more easily, and thus, the amount of Cur that can be released from the liposome is higher. 

Moreover, the drug release kinetic was studied. Many different models exist for describing drug release, some of them can be applied to nanoformulations, such as Korsmeyer-Peppas, Higuchi, Weibull, etc [51,52]. In our case, three different fitting models were performed: the Korsmeyer-Peppas model, Higuchi model, and first-order kinetic model. From the R^2^ values presented in Table 8, it can be deduced that the Korsmeyer-Peppas model was the most accurate one. This model was already described for nanoparticles [68], nanoparticles containing liposomes [69], and liposomes [70]. In addition, kinetic parameters corresponding to this model were extracted, specifically the release rate constant (K) and the release exponent (*η*). 

As was mentioned in Section 3.4.4, the *η* values that were obtained corresponded to a Fickian diffusion behavior. These values were smaller for CL, CL–HPβCD, and the ethanolic solution of Cur, and higher for the DCL–DL and Cur–HPBCD solutions. The former three samples lacked a Cur–HPβCD complex, even though CL–HPβCD contained CD (however, this CD was not complexed with any Cur). Cur–HPβCD complex is a component of the latter two samples. This indicated that HPβCD has an impact on the diffusion of Cur during drug release when the inclusion complex is formed as a result of the strong binding between the ligand and substrate.

The release rate constant (K) was also affected by the presence of CD in the formulation. Table 8 depicts that K in the HPβCD–Cur complex solution was lower than K in the ethanolic Cur solution, indicating that CD permits a slight sustained release in the solution. Contrarily, CL–HPβCD and CL showed similar constants, while DCL–DL exhibited a markedly lower one. The presence of the Cur–CD complex and its high affinity when Chems is added in liposome formulations, contribute to a more sustained release of drug.

## 5. Conclusions

DCL–DL is a novel approach to increase the encapsulation efficiency of lipophilic drugs, such as Cur, into liposomes. However, the presence of CD in the aqueous compartment can affect the integrity of the bilayer. In this work, this behavior was evaluated in terms of its elasticity, drug release, and the antioxidant activity of the drug. From these studies, we can conclude that the presence of a cyclodextrin complex in DCL–DL does not influence the elasticity of liposomes, but it does improve the antioxidant activity of Cur, and allows a more sustained release of the drug. These findings demonstrated that DCL–DL is an interesting approach to increase Cur loading into liposomes without compromising the integrity of the bilayer of deformable liposomes and enhancing Cur antioxidant activity.

## Figures and Tables

**Figure 1 pharmaceutics-10-00256-f001:**
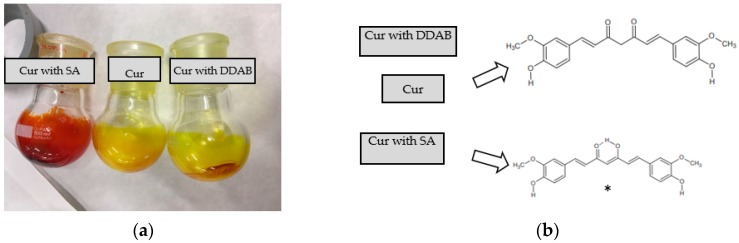
(**a**) Lipid films containing EPC, cholesterol (Cho), and Cur with an additional cationic agent: stearylamine (SA) or DDAB after removing the organic solvents methanol and chloroform. From left to right: lipid film containing Cur, Cho, and SA; lipid film containing Cur and Cho; lipid film containing Cur, Cho, and DDAB. (**b**) Chemical structures of Cur in the presence of SA or DDAB (*Source*: www.pubchem.ncbi.nlm.nih.gov).

**Figure 2 pharmaceutics-10-00256-f002:**
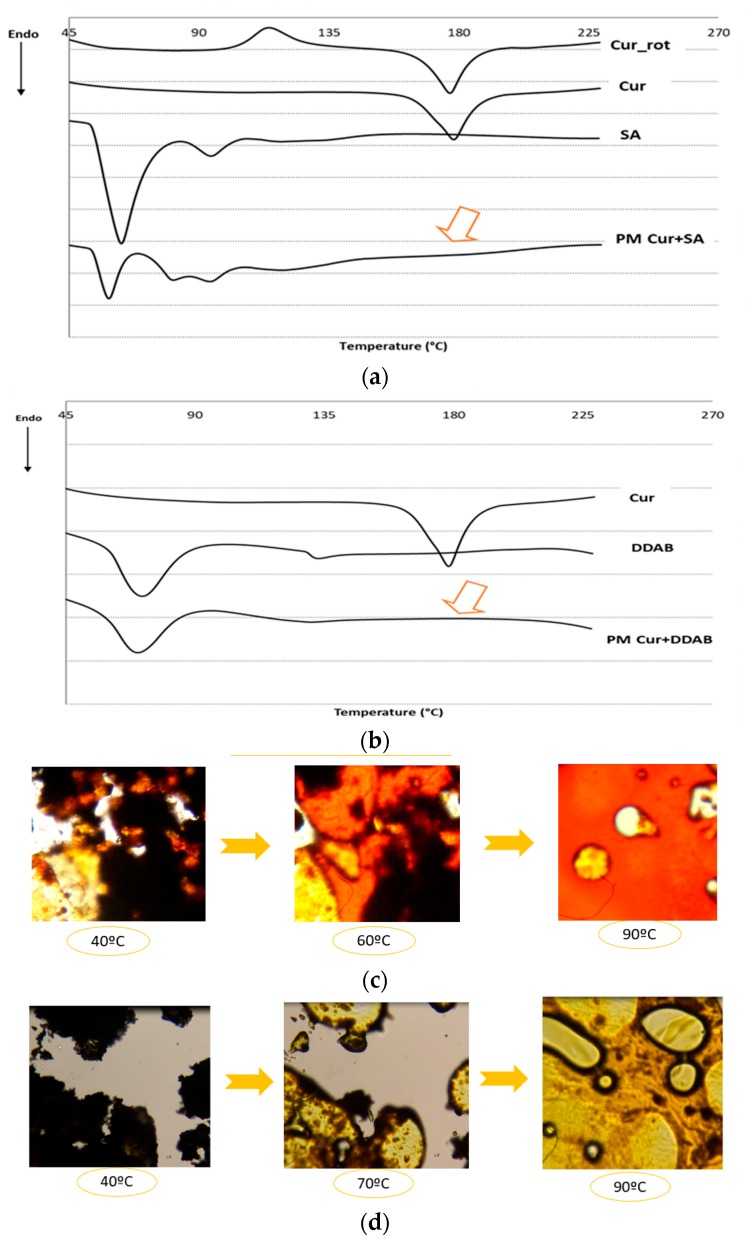
(**a**) Differential scanning calorimetry (DSC) profiles of physical mixtures of Cur with SA (PM Cur+SA) and raw materials. (**b**) DSC profiles of physical mixtures of Cur with DDAB (PM Cur+DDAB) and raw materials. (**c**) and (**d**) Hot stage microscope (HSM) images of PM Cur+SA and PM Cur+DDAB, respectively (Magnification: 400×).

**Figure 3 pharmaceutics-10-00256-f003:**
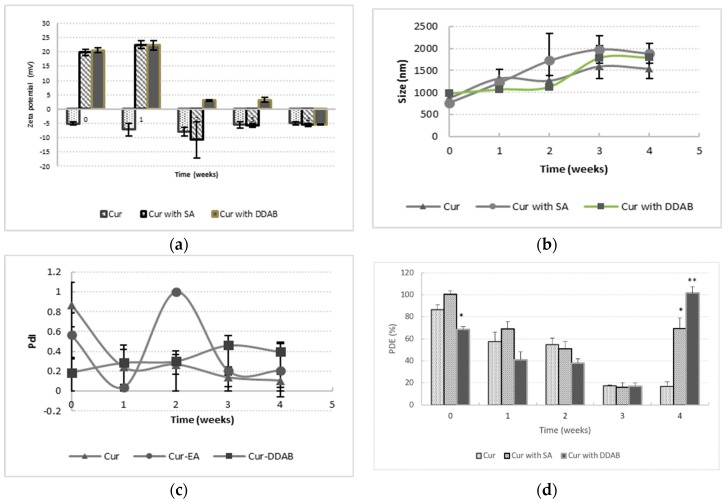
Physicochemical properties: zeta-potential (**a**), mean diameter (**b**), polydispersity index (**c**), and percentage of Cur entrapment (**d**). For each formulation, at least three different samples were prepared; values presented are mean values ± SD and were significantly different compared with the Cur group (* *p* < 0.05, ** *p* < 0.01; Student *t* test).

**Figure 4 pharmaceutics-10-00256-f004:**
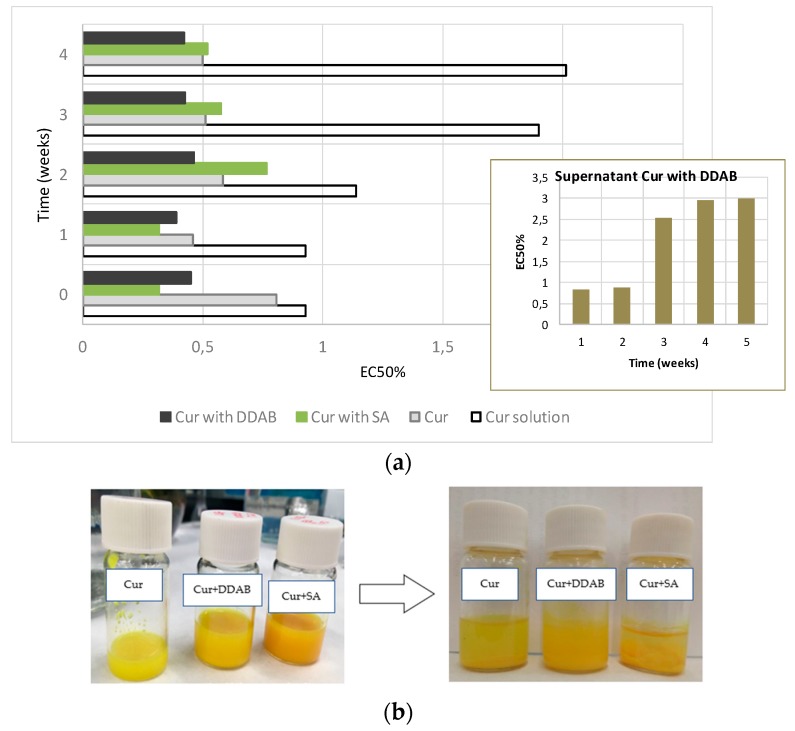
Antioxidant activity of Cur in liposome formulations with SA, DDAB, or without charged agent. (**a**) EC50% for the formulations, using Trolox as reference (EC50% 1 in red color). (**b**) Liposome samples of Cur and liposome samples of Cur containing SA or DDAB in the lipid bilayer, at time zero and after four weeks of storage.

**Figure 5 pharmaceutics-10-00256-f005:**
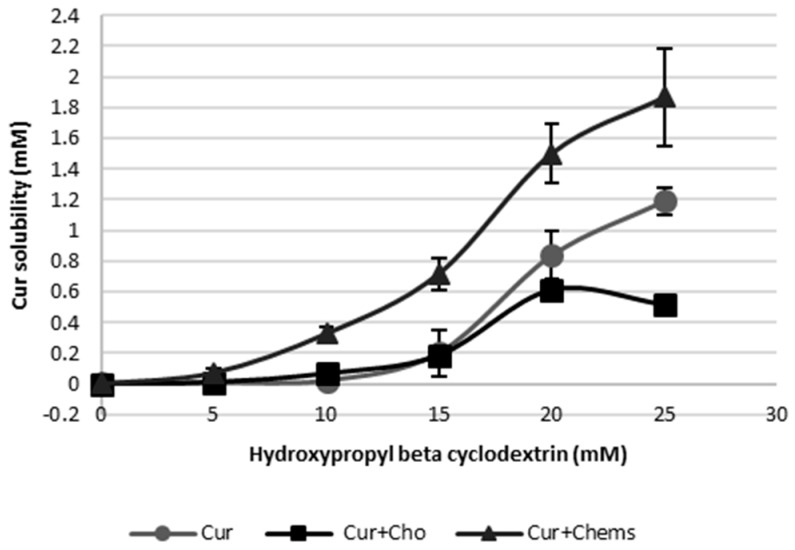
Solubility phase profiles of curcumin (Cur)–hydroxypropyl-β-cyclodextrin (HPβCD), alone and in the presence of cholesterol (Cho) or cholesteryl hemisuccinate (Chems).

**Figure 6 pharmaceutics-10-00256-f006:**
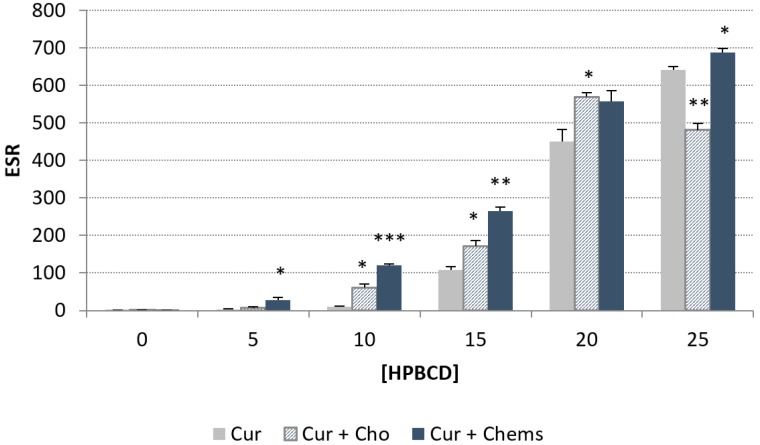
Enhancement solubility ratio (ESR) of Cur in different concentrations of HPβCD compared with Cur without CD. Values are means with standard deviation error represented by vertical bars. The mean value was significantly different compared with the Cur group at every HPβCD concentration (* *p* < 0.05, ** *p* < 0.01, *** *p* < 0.001; Student *t* test).

**Figure 7 pharmaceutics-10-00256-f007:**
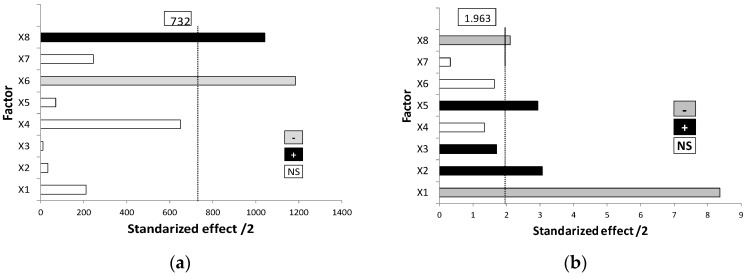
Pareto charts corresponding to: (**a**) Vesicle size. (**b**) Zeta potential. Pareto chart of the polydispersity index (PdI) is not shown because any factor was statistically significant. X1: Chems (−1 vs. +1); X2: DDAB (−1 vs. +1); X3: Cur (−1 vs. +1); X4: PL (−1 vs. +1); X5: Chems (−1, 1 vs. 0); X6: PL (−1, 1 vs. 0); X7: DDAB (−1, 1 vs. 0); X8: Cur (−1, 1 vs. 0).

**Figure 8 pharmaceutics-10-00256-f008:**
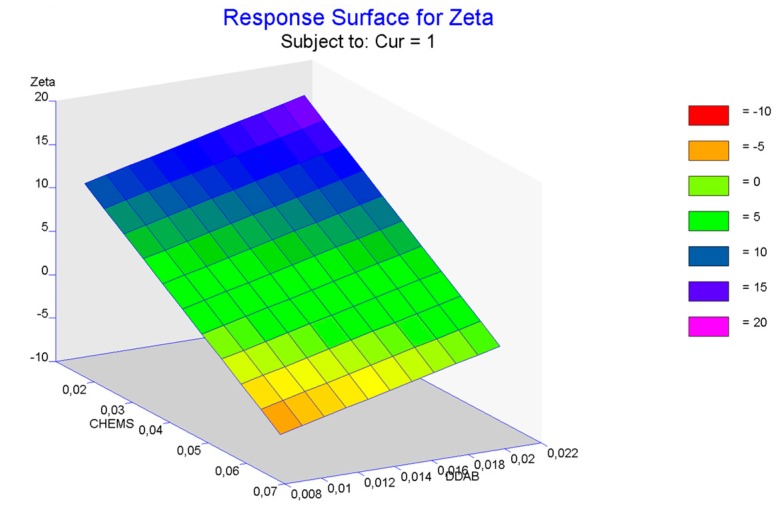
Surface plots for zeta potential (mV).

**Figure 9 pharmaceutics-10-00256-f009:**
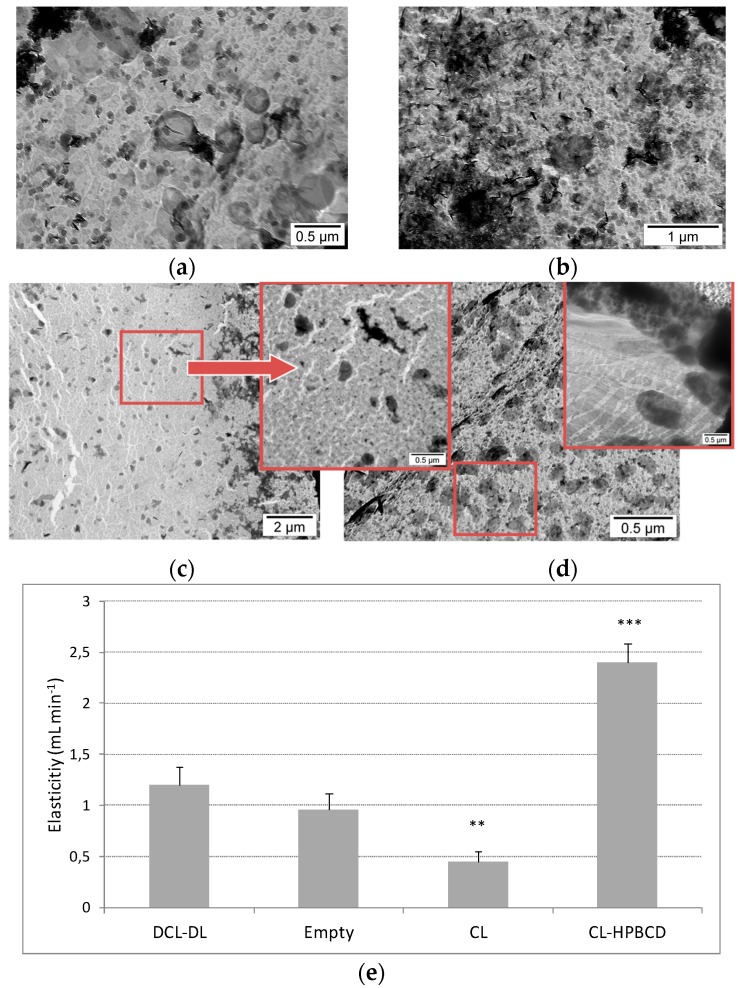
TEM images of liposomes: (**a**) and (**b**) Conventional liposomes (CL) containing buffer solution. In (**b**), it is possible to see many crystals, which is probably due to Cur precipitates. (**c**) Conventional liposomes containing 2-hydroxypropyl-α/β/γ-CD (HPBCD) in the buffer solution (CL–HPBCD). (**d**) Cur-in-cyclodextrin-in-liposomes double-loading (DCL–DL). (**e**) Elasticity of the vesicle formulations studied. Empty refers to CL without Cur. DI: deformability index. Values are means with standard deviation error represented by vertical bars. The mean value was significantly different compared with empty liposomes (** *p* < 0.01, *** *p* < 0.001; Student *t* test).

**Figure 10 pharmaceutics-10-00256-f010:**
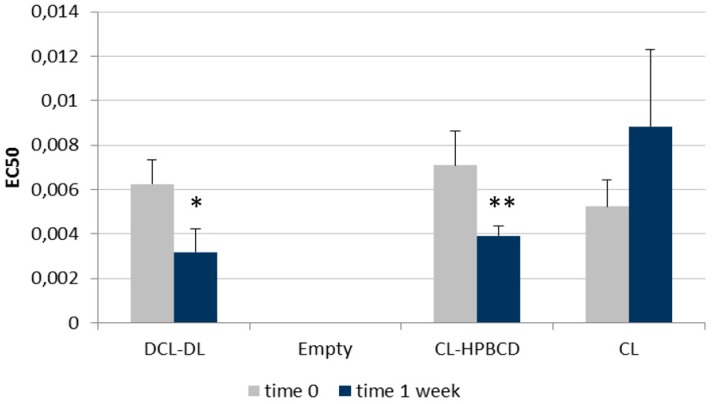
Antioxidant activity (EC50) of Cur into the different types of liposomes studied. Conventional liposomes containing buffer solution (CL). Conventional liposomes containing HPBCD in the buffer solution (CL–HPBCD). Cur-in-cyclodextrin-in-liposomes double-loading (DCL–DL). Values are means, with a standard deviation error represented by vertical bars. The mean value was significantly different compared with samples at time 0 (* *p* < 0.05, ** *p* < 0.01; Student *t* test).

**Figure 11 pharmaceutics-10-00256-f011:**
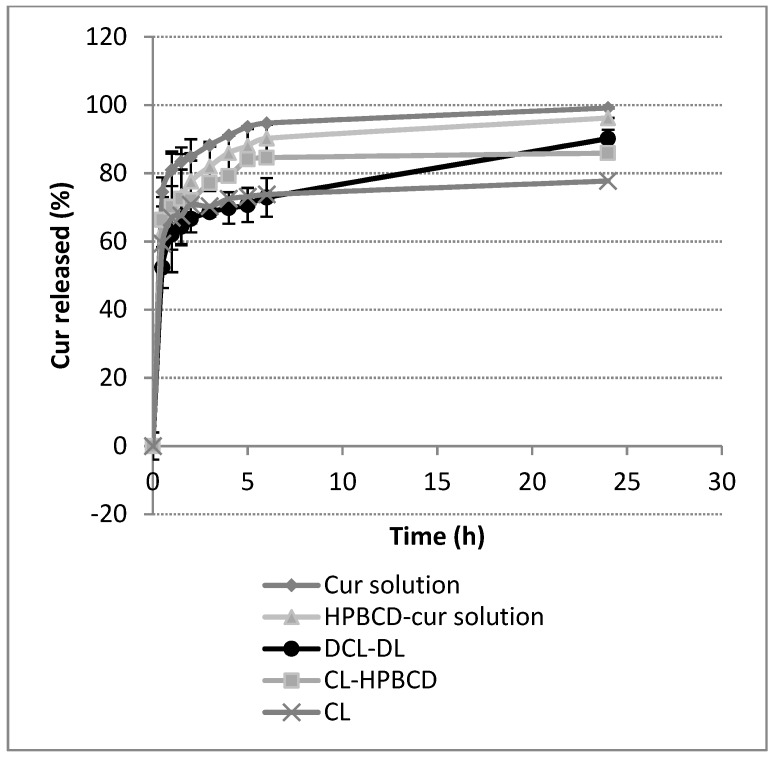
In vitro release profiles of hydroalcoholic Cur solution, aqueous solution of HPBCD–Cur complex, and DCL–DL formulation. Values are means with standard deviation error represented by vertical bar.

**Table 1 pharmaceutics-10-00256-t001:** Factors and levels selected for the experimental design. Chems: cholesteryl hemisuccinate, Cur: curcumin, DDAB: didodecyldimethylammonium bromide, EPC: l-α-Phosphatidylcholine from egg yolk.

Name	Factor	Level		
		−1	0	+1
F1	Chems (mmol)	0.0127	0.0213	0.0640
F2	DDAB (mmol)	0.009	0.015	0.021
F3	Cur (mg)	1	2	3
F4	Phospholipid (PL)	EPC	DPPC	EPC/DPPC ^1^

^1^ EPC/DPPC ratio was 1:1.

**Table 2 pharmaceutics-10-00256-t002:** Experimental matrix from L9 Taguchi array and results for size (nm), polydispersity index (PdI), and zeta potential (ZP, mV). Chems: cholesteryl hemisuccinate, Cur: curcumin, DDAB: didodecyldimethylammonium bromide, EPC: l-α-Phosphatidylcholine from egg yolk.

Exp.	Chems	DDAB	Cur	PL	Size ± SD	PdI ± SD	ZP ± SD
1	0.0127	0.009	1	Mix	3369 ± 491	0.46 ± 0.39	11.6 ± 3.1
2	0.0127	0.015	2	DPPC	2324 ± 34	0.62 ± 0.47	10.9 ± 0.6
3	0.0127	0.021	3	EPC	3838 ± 649	0.63 ± 0.52	23.8 ± 0.3
4	0.0213	0.009	2	EPC	6229 ± 1841	1 ± 0	9.2 ± 3.2
5	0.0213	0.015	3	Mix	2499 ± 284	0.91 ± 0.13	16.2 ± 4.4
6	0.0213	0.021	1	DPPC	1039 ± 130	0.54 ± 0.12	13.3 ± 0.8
7	0.064	0.009	3	DPPC	1202 ± 341	0.67 ± 0.47	−3.7 ± 5.2
8	0.064	0.015	1	EPC	3668 ± 1309	0.99 ± 0.01	1.3 ± 0.5
9	0.064	0.021	2	Mix	5267 ± 1735	0.73 ± 0.05	−1.6 ± 0.0

**Table 3 pharmaceutics-10-00256-t003:** Parameters derived from the phase solubility diagrams of Cur vs. different HPβCD concentrations. They are obtained from Equations (1)–(3) in the text.

Profile	From Equation (1)			From Equation (2)	From Equation (3)	
	*S* _0_	N (*r*^2^)	*K*_c_ (mM^−1^)	*K*_1:1_ (*r*^2^)	*K* _1:1_	*K* _1:2_
Cur	0.00185	3.72 (0.9526)	0.00424	27.97 (0.809)	4.86	0.233
Cur + Cho	0.00110	2.88 (0.9731)	0.06840	24.67 (0.809)	2	0.500
Cur + Chems	0.00271	2.082 (0.9950)	0.9530	32.041 (0.9306)	0.0295	41.278

**Table 4 pharmaceutics-10-00256-t004:** Summary of analysis of variance (ANOVA) of experimental results for vesicle size (*Y*_1_), polydispersity index (*Y*_2_), and zeta potential (*Y*_3_). X1: Chems (–1 vs. +1); X2: DDAB (–1 vs. +1); X3: Cur (–1 vs. +1); X4: PL (–1 vs. +1); X5: Chems (–1, 1 vs. 0); X6: PL (–1, 1 vs. 0); X7: DDAB (–1, 1 vs. 0); X8: Cur (–1, 1 vs. 0).

Factor	Size (*Y*_1_)		PdI (*Y*_2_)		Zeta (*Y*_3_)	
	*F*-value	*P*-value	*F*-value	*P*-value	*F*-value	*P*-value
X1	0.4218	0.532	3.1710	0.109	111.2 *	<0.001
X2	0.0115	0.917	0.0001	0.994	14.93 *	0.004
X3	0.0016	0.969	0.1052	0.753	4.555 *	0.062
X4	4.0120	0.076	0.0239	0.881	2.855	0.125
X5	0.0659	0.803	0.7078	0.422	18.20 *	0.002
X6	17.660	0.002 *	1.8470	0.207	5.646 *	0.041
X7	0.7647	0.405	0.9829	0.347	0.226	0.646
X8	13.700	0.005 *	0.9922	0.345	9.481 *	0.013

* Statistical significance (α = 0.05).

**Table 5 pharmaceutics-10-00256-t005:** Regression equations and fitting parameters for the dependent variables evaluated. MSE: mean square error.

Equation	MSE	*R*^2^ adj
Size (nm) = 2900 + 6090∙Chems	3603643	0.006
Zeta (mV) = 8.68 – 329∙Chems + 511∙DDAB + 1.69∙Cur	14.719	0.8452

**Table 6 pharmaceutics-10-00256-t006:** Optimized parameters by regression analysis.

Response	Optimized Levels F1/F2/F3/F4	Averaged Predicted Value	Confirmatory Experiment	Prediction Error
Size	−1/+1/−1/0	2978,98	3531	15.63
Zeta	−1/+1/+1/0	21,340	18.2	14.71

**Table 7 pharmaceutics-10-00256-t007:** Dissolution efficiency percentage at 6 h (%DE6h) and at 24 h (%DE24h) of release profiles. Data shown as means ± SD. The mean value was significantly different compared with CL–HPβCD liposomes (** *p* < 0.01; Student *t* test). The mean value was significantly different compared with DCL–DL liposomes (+ *p* < 0.05; Student *t*-test).

%DE	Ethanol Cur Solution	HPβCD–Cur Solution	DCL–DL	CL–HPβCD	CL
%DE6h	0.126 ± 0.001	0.116 ± 0.012	0.101 ± 0.004	0.113±0.002 ^+^	0.107 ± 0.003
%DE24h	0.497 ± 0.001	0.472 ± 0.010	0.416 ± 0.011	0.445 ± 0.005	0.406 ± 0.003 **

**Table 8 pharmaceutics-10-00256-t008:** *R*^2^ values of the first-order kinetic, Higuchi models, and Korsmeyer-Peppas models. Kinetic parameters for the Korsmeyer-Peppas model: *K* represents the release rate constant; *η* represents the release mechanism of drug.

Sample	First-order *R*^2^	Higuchi *R*^2^	Korsmeyer-Peppas *R*^2^	Log *K*	*K*	*η*
Ethanolic solution	0.5083	0.757	0.944	1.7714	59.0744926	0.0764
HPBCD–Cur complex	0.4837	0.7492	0.9346	1.6383	43.4810477	0.1182
DCL–DL	0.7285	0.9309	0.9594	1.5522	35.6615323	0.1254
CL–HPBCD	0.4514	0.684	0.8822	1.7134	51.6892225	0.0765
CL	0.4345	0.6661	0.8565	1.705	50.6990708	0.0631

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
