# Peer review of "Novel Findings about Double-Loaded Curcumin-in-HPβcyclodextrin-in Liposomes: Effects on the Lipid Bilayer and Drug Release"

_pharmaceutics, 2018, doi:10.3390/pharmaceutics10040256_

Round 1

Reviewer 1 Report

1- Title and in the text :  I suggest to change "Liposomes" by solid lipid nanoparticles

because in Figure 9 revealed only dense structure and not liposomes structure

Author Response

Thank you very much for your suggestion.

Certainly, TEM images showed similar structures to the solid state due to the desiccation of the samples during the sample processing. However, formulations proposed in this work are colloidal systems. Therefore, we will keep the title "Liposomes".

Reviewer 2 Report

In this work, Gonzalez-Rodroguez and co-works studied the influence of HPBCD-curcumin complex in the lipid bilayer properties of double-loaded liposomes. The manuscript is interesting to read and potentially useful to the field. However, the following comments should be addressed to improve the manuscript.

Figure 4b, for Vial 1, it looks like different  (volume decreased) at time zero and four weeks storage while the other vials look the same?

Table 3, the authors used a=0.1 as a cutoff for significant difference. However, a=0.05 is the common practice. The authors should clarify why they used 0.1.

Statistical analysis should be conducted in Fig. 3d, 6, 9e, and 10a.

Author Response

REVIEWER 2 (in red colour)

Thank you very much for your comments.

The introduction has been improved mainly the objectives. References have been also checked in the text.

Figure 4b, for Vial 1, it looks like different (volume decreased) at time zero and four weeks storage while the other vials look the same?

There is no reason for this. Volumes for all samples were randomly taken for pictures.

Table 3, the authors used a=0.1 as a cutoff for significant difference. However, a=0.05 is the common practice. The authors should clarify why they used 0.1.

Data have been corrected after modifying α=0.05 in the DOE software.

Statistical analysis should be conducted in Fig. 3d, 6, 9e, and 10a.

Statistical analysis has been applied to the above Figures (see in the text).

Reviewer 3 Report

The present manuscript (pharmaceutics-383549) is generally readable and the experimental studies are well conducted. Nevertheless, both in the title and in the introduction the authors have stressed on the effect of β-cyclodextrin-curcumin complex in the lipid bilayer properties of double-loaded liposomes. Actually, only elasticity of the vesicles and TEM pictures are presented regarding double-loaded liposomes without any further investigation. Please consider whether to modify the title and the aim or to discuss better the results according to the available literature (for instance:  int J Pharm. 2014 Dec 10; 476(1-2):108-15).

Line 24-27 This part of the abstract deals with release studies which are marginal in the manuscript. Please consider whether to improve release studies results or to modify the abstract. “Directly correlated” is not correct since correlation analysis was not performed.

There are some concerns regarding the references. For instance, reference 32 is missing in the introduction but it is reported in the references list. Line 98 and Line 154: references are reported in a different format (Chiang et al., 2014) instead of [1].

Line 345 What is shape 1 about melting profile of Curcumin?

Legend in figure 1a is incomplete.

Figure 4 For what reason rectangles were drawn in the picture?

Conclusion should reflect the findings of the manuscript in relation to the aim of the work.

Author Response

REVIEWER 3 (in green colour)

Moderate English changes required 

An exhaustive revision of the manuscript has been carried out.

Both in the title and in the introduction the authors have stressed on the effect of β-cyclodextrin-curcumin complex in the lipid bilayer properties of double-loaded liposomes. Actually, only elasticity of the vesicles and TEM pictures are presented regarding double-loaded liposomes without any further investigation. Please consider whether to modify the title and the aim or to discuss better the results according to the available literature (for instance:  int J Pharm. 2014 Dec 10; 476(1-2):108-15).

The title has been changed to Novel findings about double-loaded curcumin-in HPbcyclodextrin-in liposomes: effects on the lipid bilayer and drug release”. In addition, results and discussion of in vitro release studies have been significantly improved.

Line 24-27 This part of the abstract deals with release studies which are marginal in the manuscript. Please consider whether to improve release studies results or to modify the abstract.

Release studies have been significantly improved (see in the text)

 “Directly correlated” is not correct since correlation analysis was not performed.

Abstract has been reviewed according to suggested changes by reviewers.

There are some concerns regarding the references. For instance, reference 32 is missing in the introduction but it is reported in the references list. Line 98 and Line 154: references are reported in a different format (Chiang et al., 2014) instead of [1].

All references have been checked in the manuscript.

Line 345 What is shape 1 about melting profile of Curcumin?

Sorry, we wanted to say that “the shape of this profile corresponds to Form I of the different polymorphs that Cur has, the most usual shape” (see in the text)

Legend in figure 1a is incomplete.

This legend has been changed in Figure 1a (see in the text).

Figure 4. For what reason rectangles were drawn in the picture?

Erroneously, the inner text was missed from rectangles. It has been added (see in the text).

Conclusion should reflect the findings of the manuscript in relation to the aim of the work.

Conclusions have been adequately changed according the aim of the work.

Round 2

Reviewer 2 Report

Ready to publish.

Author Response

Thank you very much for your review

Reviewer 3 Report

The manuscript is quite improved after the revision.

Only few comments:

Line 97 Merge references into one bracket.

Remove the year in the brackets after the expression author et al in the manuscript

AUC (area under curve) is generally referred for in vivo bioavailability. Please, replace AUC with another expression in the definition of the in vitro dissolution efficiency.

The conclusions sounds as an abstract. They don’t point out briefly and in a concise way the message of the paper.

Author Response

Line 97 Merge references into one bracket.

All separate references have been merged

Remove the year in the brackets after the expression author et al in the manuscript

The year in the brackets has been removed in lines 83, 134, 262, 323, 614, 774, 804 and 823.

AUC (area under curve) is generally referred for in vivo bioavailability. Please, replace AUC with another expression in the definition of the in vitro dissolution efficiency.

AUC term has been replaced by area under the dissolution curve up to time t (AUDC0t).

The conclusions sounds as an abstract. They don’t point out briefly and in a concise way the message of the paper.

The conclusions have been changed. The paragraph results as follows:

“DCL-DL are a novel approach to increase encapsulation efficiency of lipophilic drugs, such as Cur, into liposomes. However, the presence of CD in the aqueous compartment can affect the integrity of the bilayer. In this work, this behavior was evaluated in terms of elasticity, drug release and antioxidant activity of drug. From these studies, we can conclude that the presence of cyclodextrin complex in DCL-DL do not influence the elasticity of liposomes, improves the antioxidant activity of Cur and allows a more sustained release of drug. These findings demonstrated that DCL-DL are an interesting approach to increase Cur loading into liposomes without compromising the integrity of the bilayer of deformable liposomes and enhancing Cur antioxidant activity”.